# Conformation, Self-Organization and Thermoresponsibility of Polymethacrylate Molecular Brushes with Oligo(ethylene glycol)-block-oligo(propylene glycol) Side Chains

**DOI:** 10.3390/polym13162715

**Published:** 2021-08-13

**Authors:** Maria Simonova, Denis Kamorin, Oleg Kazantsev, Maria Nepomnyashaya, Alexander Filippov

**Affiliations:** 1Institute of Macromolecular Compounds of the Russian Academy of Sciences, Bolshoy Prospekt 31, 199004 Saint Petersburg, Russia; afil@imc.macro.ru; 2Laboratory of Acrylic Monomers and Polymers, Department of Chemical Technology, Dzerzhinsk Polytechnic Institute, Nizhny Novgorod State Technical University n.a. R.E. Alekseev, 24 Minin Street, 603950 Nizhny Novgorod, Russia; d.kamorin@mail.ru (D.K.); kazantsev@dpingtu.ru (O.K.); 3Chromatography Laboratory, Department of Production Control and Chromatography Methods, Lobachevsky State University of Nizhni Novgorod, Dzerzhinsk Branch, 23 Prospekt Gagarina, 603950 Nizhny Novgorod, Russia; 4Higher School of Technology and Energy, Ivana Chernykh 4, 198095 Saint Petersburg, Russia; marinepom@mail.ru

**Keywords:** molecular brushes, radical polymerization, thermoresponsive polymers, methoxy (oligoethylene glycol-block-oligopropylene glycol) methacrylates, oligo(ethylene glycol) methacrylates, oligo(propylene glycol) methacrylates

## Abstract

Polymethacrylic molecular brushes with oligo(ethylene glycol)-block-oligo(propylene glycol) side chains were investigated by static and dynamic light scattering and viscometry. The solvents used were acetonitrile, tetrahydrofuran, chloroform, and water. The grafted copolymers were molecularly dispersed and dissolved in tetrahydrofuran and acetonitrile. In these solvents, the molar masses of copolymers were determined. In thermodynamically good solvents, namely tetrahydrofuran and acetonitrile, investigated copolymers have a high intramolecular density and the shape of their molecules resembles a star-shaped macromolecule. In chloroform and water, the micelle-like aggregates were formed. Critical micelle concentrations decreased with the lengthening of the hydrophobic block. Molecular brushes demonstrated thermosensitive behavior in aqueous solutions. The phase separation temperatures reduced with an increase in the content of the oligo(propylene glycol) block.

## 1. Introduction

Polymers with complex architecture attract a significant amount of attention due to the wide possibilities of regulating their characteristics by the variation of the structure and the architecture parameters [1,2]. In the case of molecular brushes, these parameters are the chemical structure of the backbone and side chains, their sizes, and the grafting density of the grafted chains. The introduction of functional groups into the main and/or side chains is also used. The modern synthetic approaches make it possible to obtain grafted copolymers with an ordained molecular architecture and controlled molar mass.

Thermoresponsive molecular brushes are of special interest because they are used in various fields—for example, in medicine and biotechnology, in solving environmental problems, etc. [3,4] A significant number of molecular brushes are amphiphilic systems since they are built from components with different chemical natures. Therefore, they may adopt a wide variety of conformations in selective solvents [5,6]. The behavior of their solutions is influenced importantly by the different thermodynamic qualities of the solvent being used with respect to the backbone and side chains. Moreover, the architecture parameters of grafted copolymers strongly affect the solubility and the assembly behavior of macromolecules compared to linear diphilic block-copolymers [7,8,9,10]. For example, the shape of the molecules of dense amphiphilic brushes in a thermodynamically poor solvent for the backbone resembles a star-shaped molecule, the core of which is a collapsed main chain, and the grafted chains are arms [11,12,13,14] A decrease in the grafting density of the side chains leads to a decline in solubility and to aggregate formation due to the interaction of insoluble components [15,16]. In particular, diphilic molecular brushes are able to form ordered micellar structures by the self-assembly processes of macromolecules in selective solvents.

A large number of works are devoted to the analysis of the stimulus sensitivity of molecular brushes [12,13,17]. However, the number of systematic studies of the concrete classes of polymer brushes is not very large. The effects of the structure of components, their fraction, grafting density, and molar masses on self-organization and phase separation temperatures are analyzed. The role of side chain grafting density z in the formation of the properties of aqueous solutions was established for the grafted copolymers with an alkylene-aromatic polyester backbone and side poly(2-ethyl-2-oxazoline) chains [18]. The decrease in z facilitates the contacts of the backbones of different macromolecules. In the solution of copolymer with low z, the aggregate dimensions were larger than those in the case of brushes with high grafting density. The phase separation temperature reduced with a decrease in z. Moreover, at low temperatures, the aggregation is the dominating process in the solution of low grafting density copolymers, while for brushes with high z, the compaction of macromolecules and aggregates prevailed.

Amphiphilic copolymers containing methoxy oligo(ethylene glycol) methacrylates (MOEGM) have been intensively studied over the last 20 years [19,20]. Biocompatible MOEGM polymers are considered to be drug delivery polymers [19,20,21,22,23,24]. The influence of the structure of such copolymers on solution behavior, self-organization, and association have been investigated [25,26,27,28]. Recently, it has been shown that copolymers of MOEGM with higher alkyl methacrylates formed unimolecular micelles in aqueous solutions [29,30,31,32,33]. The thermoresponsive behavior of this type of polymers has also been demonstrated. The synthesis of polymer molecular brushes based on diblock macromonomers (methoxy (oligo(ethylene glycol)_e_-block-oligo(propylene glycol)_p_) methacrylates, OEGeOPGpMA) with different lengths of oligo(ethylene glycol) (e) and oligo(propylene glycol) (p) moieties was reported recently [34]. The effects of synthesis conditions on the polymerization rates, maximum conversions, molar masses, and polydispersity have been analyzed.

The aim of this work is to investigate the effect of oligo(ethylene glycol) and oligo(propylene glycol) block lengths in polymers on the molecular and hydrodynamic characteristics of molecular dispersity solutions and the behavior of aqueous solutions (self-organization and thermoresponsibility) of polymethacrylic molecular brushes with oligo(ethylene glycol)-block-oligo(propylene glycol) side chains. The structures of the investigated polymers are presented in Figure 1.

## 2. Materials and Methods

### 2.1. Copolymer Synthesis

The “grafting through” method was used to produce polymers with a brush structure. The synthesis of the polymer brushes by the “grafting through” method implies a one-step process using macromonomers capable of radical polymerization. Three macromonomers with different oligo(ethylene glycol) (OEG) and oligo(propylene glycol) (OPG) block lengths were used to obtain polymers (Dzerzhinsk Polytechnic Institute, Dzerzhinsk, Russia; Sigma-Aldrich, Saint Louis, MO, USA). We used OEGeOPGpMA with the following average lengths of oligo(ethylene glycol) (e) and oligo(propylene glycol) (p) fragments: e = 7.0 and p = 2.8 in copolymer E7P3, e = 7.0 and p = 5.4 in brush E7P5, and e = 7.0 and p = 10.3 in sample E7P10 (Dzerzhinsk Polytechnic Institut, Dzerzhinsk, Russia). Polymers were synthesized by the conventional free-radical polymerization in organic solvents at 60–85 °C. The synthesis procedure of OEGeOPGpMAs was described in detail previously [34].

The structure of the polyOEGeOPGpMA samples was confirmed by nuclear magnetic resonance (NMR) spectroscopy (DDR2 400; Agilent, Santa Clara, CA, USA) in DMSO-D6 (Appendix A).

### 2.2. Methods Molecular Hydrodynamics and Optics

The initial characterization of the OEGeOPGpMA copolymers was carried out by gel permeation chromatography using a Chromos LC-301 (Chromos Engineering Co. Ltd., Dzerzhinsk, Russia) instrument with isocratic pump, refractometric detector and two exclusive columns Phenogel 5u 50A (Phenomenex, Torrance, CA, USA). Tetrahydrofuran (THF) was used as a mobile phase. Polystyrene calibration was used to calculate the molar masses of polymers, Appendix A.

The solutions of polyOEGeOPGpMA (3 samples) were studied by molecular hydrodynamics and optics. The static (SLS) and dynamic light scattering (DLS) experiments were carried out using a Photocor Complex instrument (Photocor Instruments Inc., Moscow, Russia). The light source was the Photocor-DL diode laser with the wavelength λ = 659.1 nm and controllable power up to 30 mW. The correlation function of the scattered light intensity was obtained using the Photocor-PC2 correlator with 288 channels and processed using the DynalS software (ver. 8.2.3, SoftScientific, Tirat Carmel, Israel). Toluene was used as a calibration liquid, whose absolute scattering intensity is equal to 1.38 × 10^−5^ cm^−1^. The measurements were performed at scattering angles θ in the range 45–135°.

Chloroform (density ρ_0_ = 1.486 g∙cm^−3^, dynamic viscosity η_0_ = 0.57 cP, and refractive index *n*_0_ = 1.443), TGF (ρ_0_ = 0.890 g∙cm^−3^, η_0_ = 0.46 cP, and *n*_0_ = 1.405), acetonytryle (ρ_0_ = 1.486 g∙cm^−3^, η_0_ = 0.57 cP, and *n*_0_ = 1.443), and water (ρ_0_ = 1.000 g∙cm^−3^, η_0_ = 0.98 cP, and *n*_0_ = 1.333) were used as solvent (Sigma-Aldrich, Saint Louis, MO, USA). The values were determined at 25 °C.

For all solutions, the DLS recorded a unimodal particle size distribution, i.e., there was one type of particle in all of the solvents (Figure 2). This behavior is in agreement with the chromatographic data (Appendix A). The values of the hydrodynamic radii *R*_h-D_(*c*) at given concentration *c* were determined in the wide concentration range and extrapolated to zero concentration to obtain the hydrodynamic radius *R*_h-D_ of macromolecules (Appendix A). The translation diffusion coefficients *D*_0_ and friction coefficient *f* were obtained using Stokes–Einstein equations:*D*_0_ = *k*_B_*T*/*f* = *k*_B_*T*/(6πη_0_*R*_h-D_)(1)
where *k*_B_ is Boltzmann’s constant and *T* is absolute temperature.

In all solutions, the asymmetry of the light scattering intensity was not observed for all copolymers. Therefore, the gyration radii of scattering objects could not be determined, and polymer molar masses *M*_w_ were obtained by the Debye method (angle 90°), using the formula
(2)$$\frac{cH}{I_{90}}=\frac{1}{M_{w}}+2A_{2}c$$
where *A*_2_ is second virial coefficient and *H* is the optical constant:(3)$$H=\frac{4\pi^{2}n_{{}_{0}}^{2}{\left(dn/dc\right)}^{2}}{N_{A}\lambda_{0}^{4}}$$

Here, *I*_90_ is the excessive intensity of light scattered at an angle of 90°, *N*_A_ is Avogadro’s number, and *dn*/*dc* is the refractive index increment. The values of *dn*/*dc* were determined using an RA-620 refractometer (KEM, Kyoto, Japan) with a wavelength *λ*_0_ = 589.3 nm. The values of *dn*/*dc* were calculated from the slope of concentration dependences of the difference Δ*n* = *n − n*_0_ between the refractive indexes of the solution *n* and the solvent *n*_0_. The Debye plots for the investigated solutions are shown in Appendix A. Note that acetonitrile was thermodynamically good for all polymer samples (the second virial coefficient was positive). The values *M*_w_ and *R*_h-D_ are shown in Table 1.

The viscosimetry experiments were carried out on the Ostwald-type Cannon-Manning capillary viscometer (Cannon Instrument Company Inc., State College, PA, USA). The efflux time of the solvent was 104.5 s. The dependencies of the reduced viscosity η_sp_/*c* on the concentration were analyzed using the equation of Huggins:η_sp_/*c* = [η] + *k*_H_[η]^2^*c*(4)
where [η] is the intrinsic viscosity and *k*_H_ is the Huggins constant.

The solutions, solvent, and calibration liquid were filtered into cells that were ensured to be dust-free previously by benzene. Millipore filters (Millipore Corp., Billerica, MA, USA) with a PTFE membrane with the pore size of 0.20 μm were used. The described experiments were carried out at 21 °C. Before the experiments, the scattering cells were rinsed with benzene, evacuated for 15 min, and filled with dust-free air. The solutions were prepared at room temperature. All solutions were stored for at least 12 h prior to measurements in order to ensure a complete equilibration.

The critical micelle concentrations (CMC) of copolymers were determined by fluorimetry using pyrene as a fluorescent probe [35,36]. The aqueous solutions of polymers with 10 different concentrations in the range of 1∙10^−6^ ÷ 0.5 mg/mL were prepared by dissolving the polymers in aqueous pyrene solutions (6∙10^−7^ M). The resulting mixtures were then sonicated for 5 min and allowed to equilibrate for 24 h at room temperature before measurements. Steady-state fluorescence spectra were recorded on a Shimadzu RF-6000 spectrofluorimeter (Shimadzu, Kyoto, Japan) under the following conditions: the excitation slit width was 3 nm, the emission slit width was 3 nm, the scanning speed was 200 nm/min, the excitation wavelength was 335.0 nm, the emission wavelength was 350.0–500.0 nm, and the temperature was 25 °C. The ratio between the intensities (*I*_1_/*I*_3_) of the first (*I*_1_, 373 nm) and the third (*I*_3_, 384 nm) vibronic bands of pyrene emission were plotted against the polymer concentration (Figure 3). Critical micelle concentrations were determined as a concentration corresponding to the inflection point at which *I*_1_/*I*_3_ started to decrease.

### 2.3. Investigation of the Self-Assembly of PolyOEGeOPGpMA in Water Solutions

The processes of the self-organization of polyOEGeOPGpMA in aqueous solutions were investigated by the methods of s SLS, DLS, and turbidimetry using the Photocor Complex (Photocor Instruments Inc., Moscow, Russia) described above, which was also equipped with the Photocor-PD detection device for measuring the transmitted light intensity. The experimental procedure was described in detail previously [37]. The polymer concentration was equal to 0.005 g∙cm^−3^ for all copolymer samples. The solution temperature *T* was changed discretely with the step ranging from 1.0 to 5.0 °C. The temperature was regulated with the precision of 0.1 °C. The *T* values changed in a range from 15 to 79 °C. After a given temperature was achieved, all of the experimental characteristics began to change in time and reached constant values in time *t*_eq_. At steady-state conditions, i.e., when the solution characteristics do not depend on time, the intensity *I* of scattered light, optical transmition *I**, hydrodynamic radii *R*_h_ of the scattering species, and their contribution *S*_i_ to the integral scattering intensity were determined. *S*_i_ was estimated using the values of the areas under the curved line of the corresponding *R*_h_ distribution peak. These measurements were performed at a scattering angle range from 45° to 135° to prove the diffusion nature of the modes. To maintain the linearity of the instrument with respect to *I*, the amount of fixed light scattering was attenuated by filters and by reducing the laser power so that the measured value of *I* did not exceed 1.2 MHz.

## 3. Result and Discussion

### 3.1. Molar-Masses and Hydrodynamic Characteristics

For all samples of polyOEGeOPGpMA, the molar masses, hydrodynamic radii of the macromolecules, and intrinsic viscosities determined by the static light scattering method (SLS) (Photocor Instruments Inc, Moscow, Russia) and Gel permeation chromatography (GPC) (Chromos LC-301, Chromos Engineering Co. Ltd., Dzerzhinsk, Russia) instrument with isocratic pump, refractometric detector and two exclusive columns Phenogel 5u 50A (Phenomenex, Torrance, CA, USA) in tetrahydrofuran (THF) and acetonitrile solutions coincide within the experimental error (Table 1). Using the obtained *M*_w_ values, it is easy to calculate the polymerization degree *N*_b_ of the backbone of the copolymers according to the equation
*N*_b_ = *M*_w_/*M*_0-cp_(5)
where *M*_0-cp_ is molar masses repeating the units of OEGeOPGpMA: *M*_0-cp_ = 582 g∙mol^−1^ for polyE7P3, *M*_0-cp_ = 688 g∙mol^−1^ for polyE7P5, and *M*_0-cp_ = 988 g∙mol^−1^ for polyE7P10. The *N*_b_ values are listed in Table 2. Table 2 also presents the average values of the length *L*_b_ = *N*_b_ λ_0-b_ of backbone. Length *L*_b_ was calculated under the assumption that all valence bonds have the same length of 0.14 nm. The valence angles are tetrahedral and, consequently, the length of the repeating unit of the main chain λ_0-b_ = 0.25 nm. The lengths *L*_sc_ of the side chains were determined in the same way (Table 2).

It is clearly seen that, for investigated samples, the lengths of the main and side chains are close. Back in 1986, V. Tsvetkov suggested that comb-like macromolecules with such a ratio of length components, even in a good solvent, should have a shape similar to a star-shaped macromolecule [11]. Simplified schemes of the molecules of the studied samples are shown in Figure 4. It is convenient to model such structures with an ellipsoid of revolution [11]. Taking into account the *L*_b_ and *L*_sc_ values, it can be assumed with great confidence that the ratio of the length and short axes of the modeling ellipsoids for the studied polymers will not differ very much from unity.

The fact that the shape of polyOEGeOPGpMA molecules resembles a star-shaped macromolecule is evidenced by the obtained low values of intrinsic viscosity (Table 1). Such low values are typical for multi-armed polymer stars [38,39]. They indicate the compact size and symmetrical shape of macromolecules. The compact size and increased intermolecular density are confirmed by the low values of the hydrodynamic radii *R*_h-D_ determined in acetonitrile and THF (Table 1). In addition, for the studied copolymers, low values of the hydrodynamic variant *A*_0_ were obtained (Table 1). *A*_0_ were calculated using the experimental values of the values of *M*_w_ [η] and *D*_0_ by the formula [11,40,41]
(6)$$A_{0}=\eta_{0}D_{0}{\left(\frac{M\left[\eta \right]}{100}\right)}^{1/3}/T_{a}$$
where *T*_a_ is absolute temperature. As known, the average theoretical and experimental values of the hydrodynamic invariant for flexible-chain polymers is *A*_0_ = (3.2 ± 0.2)·10^−10^ erg·K^−1^mol^−1/3^ [11,40,41], and the theoretical estimate for the impermeable solid sphere gives *A*_0_ = 2.88·10^−10^ erg·K^−1^mol^−1/3^. For multi-armed polymer stars, a decrease in the experimental values of *A*_0_ was observed in comparison with *A*_0_ for linear polymers. For example, for six-arm poly (2-alkyl-2-oxazolines) *A*_0_ = 2.8·10^−10^ erg·K^−1^mol^−1/3^ [42]. The *A*_0_ value for star-shaped polymers decreases with an increase in the number of arms and a decrease in their length.

Thus, the performed analysis suggests that, in tetrahydrofuran and acetonitrile, the conformation of the studied copolymers is similar to the conformation of a polymer star, which has an elongated core (main chain) and whose arms are side chains. The number of the latter coincides with the polymerization degree of the backbone.

On passage from tetrahydrofuran and acetonitrile to chloroform and water, a change in the hydrodynamic characteristics was observed (Table 1). The intrinsic viscosities [η] increased most strongly. The values of the hydrodynamic radius *R*_h-D_ and hydrodynamic invariant *A*_0_ behaved in exactly the same way. Their changes were not very great, but it can be traced reliably. It can be assumed that the aggregates were formed in these solvents, which is reflected in the experimental values of the molar masses (Table 1).

Aggregate formation is confirmed by fluorimetry. Table 1 presents the critical micelle concentrations of brushes in water. It was found that polymers have CMC values of about 3∙10^−4^ wt.%. Considering the low CMC values, it is obvious that the sizes determined by DLS in water are the sizes of polymer micelles. Furthermore, given the similar molar masses of the samples, one can observe a clear tendency for the CMC to decrease (from 3.35·10^−4^ to 2.70∙10^−4^) as the length of the hydrophobic blocks in the side chains increases.

### 3.2. Properties of the Aqueous Solutions of PolyOEGeOPGpMA

At room temperature, the dynamic light scattering method recorded the existence of only one mode in aqueous solutions of the investigated samples. As well as for solutions of polyOEGeOPGpMA in acetonitrile and THF, the hydrodynamic radius *R*_h-m_ = *R*_h-D_ of particles corresponding to this mode was determined by the extrapolation to zero concentration (Appendix A). The values of radius *R*_h-m_ depended on the length of the hydrophobic block in the side chains. The highest aggregate sizes were obtained for polyE7P10 (*R*_h-m_ = 4.3 nm), the smallest for polyE7P3 (*R*_h-m_ = 3.2 nm), and *R*_h-m_ = 4.2 nm for polyE7P5.

The phase transition was detected in polyOEGeOPGpMA solutions upon heating by light scattering and turbidimetry methods. Figure 5 shows the temperature dependences of the relative optical transmission *I**/*I**_21_ for the studied solutions. The observed behavior is typical for thermoresponsive polymers [43,44,45,46,47,48]. The temperatures of the onset *T*_1_ and the finishing *T*_2_ of the phase separation were defined as the beginning of the decrease in *I**/*I**_21_ and as reaching its minimum value, respectively. With the elongation of the hydrophobic component in the side chains of polyOEGeOPGpMA, the dependences *I**/*I**_21_(*T*) shift to the region of low temperatures, i.e., the phase separation temperatures *T*_1_ and *T*_2_ decrease with increasing OPG content.

At *T* < *T*_1_, the light scattering intensity *I* did not change upon heating (Figure 6). This is explained by the independence of the radii of scattering objects *R*_h-m_ on temperature (Figure 7 and Appendix A). Within the phase separation interval, the strong increase in intensity *I* was observed for all solutions. Such behavior was caused by the change in the composition of the scattering species and their dimensions. Near the temperature of onset *T*_1_ of the phase separation, the new aggregates with hydrodynamic radii *R*_h-s_ appeared in the solutions. The main reason for the formation of these supramolecular structures is the dehydration of the OPG block. For all samples, the radius increased upon heating and reached the maximal value at temperature *T*_2_, at which point the intensity of scattered light became maximum. Near *T*_1_, micelle-like aggregates with a radius of *R*_h-m_ cease to be detected by the dynamic light scattering since they formed large aggregates with a radius *R*_h-s_.

At *T* > *T*_2_, both *I* and *R*_h-m_ decreased with temperature. This fact can reflect the compaction of the copolymer molecules and aggregates due to the dehydration of side chains. However, the solutions were turbid and the light scattering was not classical (multiple scattering) [49,50].

## 4. Conclusions

Molecular brushes polyOEGeOPGpMA were investigated by the methods of molecular hydrodynamics and optics in dilute solutions in acetonitrile, THF, chloroform, and water. The polymerization degree of the studied samples was about 20. The copolymers are molecularly dispersedly dissolved in THF and acetonitrile. In these solvents, OEGeOPGpMA are characterized by a high intramolecular density, and the shape of their molecules resembles a star-shaped macromolecule. The core of this “star” is the extended backbone of the grafted copolymer, and the arms are the side chains. In chloroform and water, the formation of small micelle-like aggregates was observed. CMC in water decreased with the lengthening of the hydrophobic OPG block.

In aqueous solutions, the studied copolymers demonstrated thermoresponsibility. The phase separation temperatures decreased with an increase in the content of the OPG block in the side chains. For all studied copolymers, phase separation occurred with the formation of large micron-sized aggregates.

## Figures and Tables

**Figure 1 polymers-13-02715-f001:**
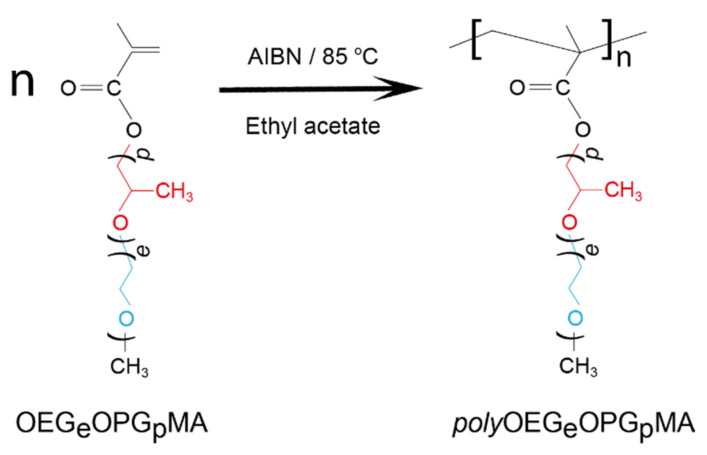
Structures of the studied polymer brushes.

**Figure 2 polymers-13-02715-f002:**
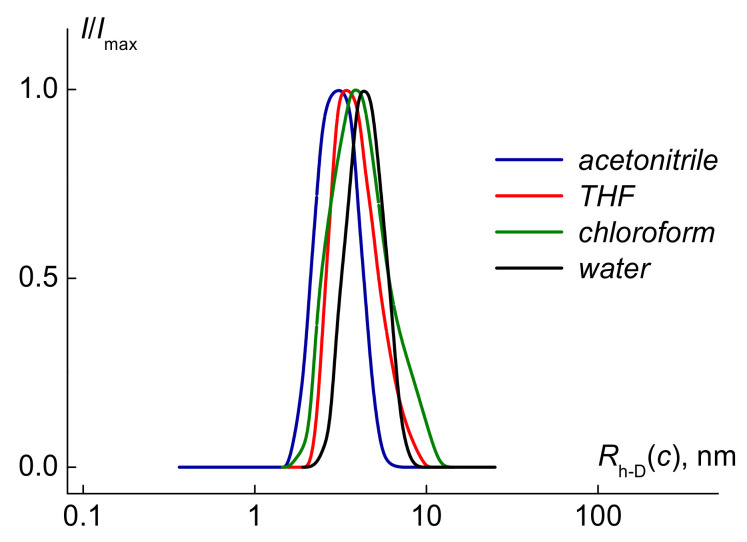
The hydrodynamic radii distribution for the solution of polyE7P10 at concentration c ≈ 0.005 g·cm^−3^. *I*_max_ is the maximum intensity of scattered light for the given solution concentration.

**Figure 3 polymers-13-02715-f003:**
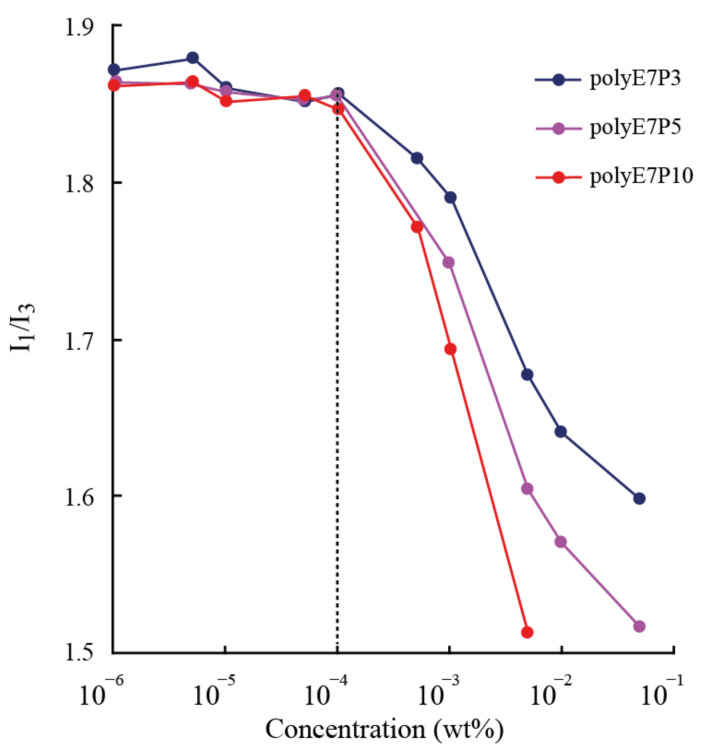
*I*_1_/*I*_3_ ratio of the vibronic band intensities of pyrene as a function of a polymer concentration.

**Figure 4 polymers-13-02715-f004:**
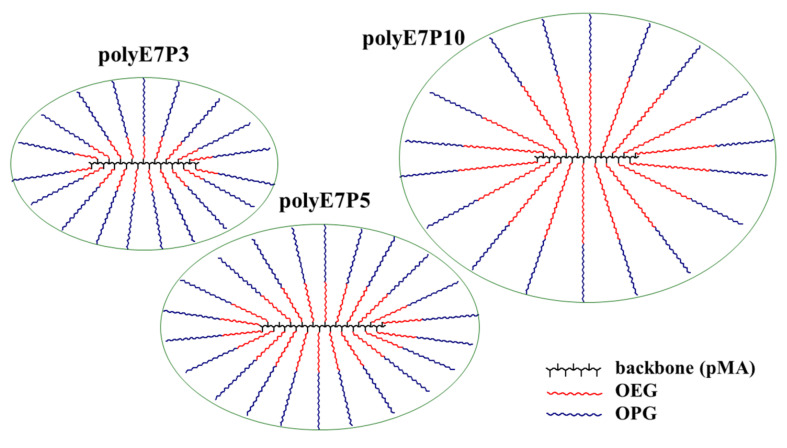
Simplified schemes of the molecules of polyOEGeOPGpMA.

**Figure 5 polymers-13-02715-f005:**
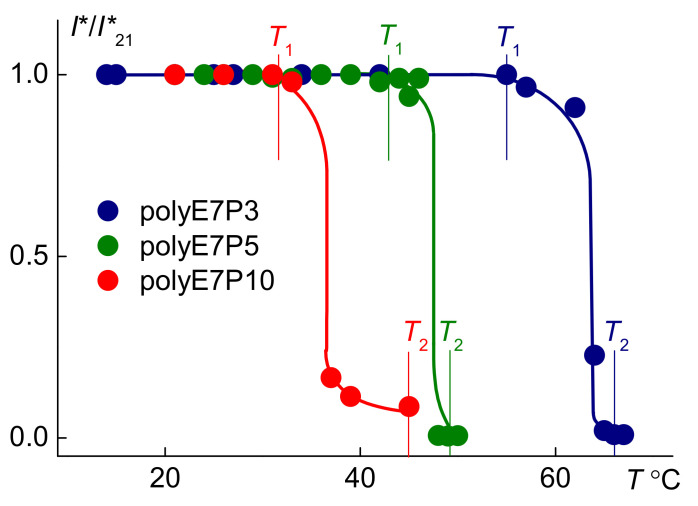
Temperature dependences of the relative optical transmission *I**/*I**_21_ for the aqueous solutions of polyOEGeOPGpMA at concentration *c* ≈ 0.0050 g·cm^−3^*. I**_21_ is optical transmission at 21 °C.

**Figure 6 polymers-13-02715-f006:**
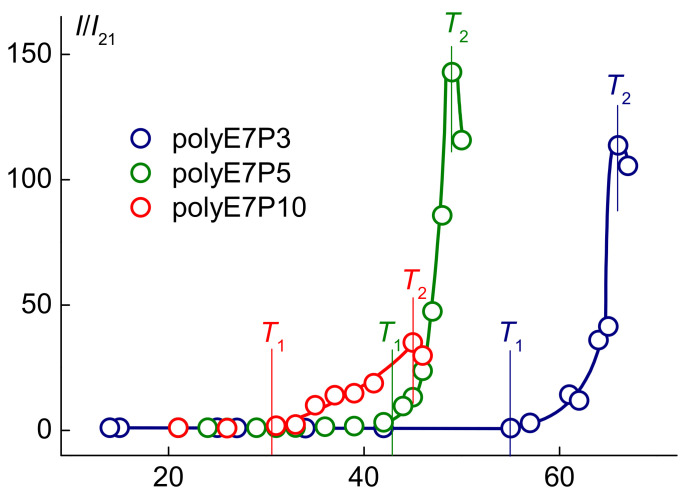
Temperature dependences of the relative light scattering intensity *I*/*I*_21_ for the aqueous solutions of polyOEGeOPGpMA at concentration *c* ≈ 0.0050 g·cm^−3^*. I*_21_ is light scattering intensity at 21 °C.

**Figure 7 polymers-13-02715-f007:**
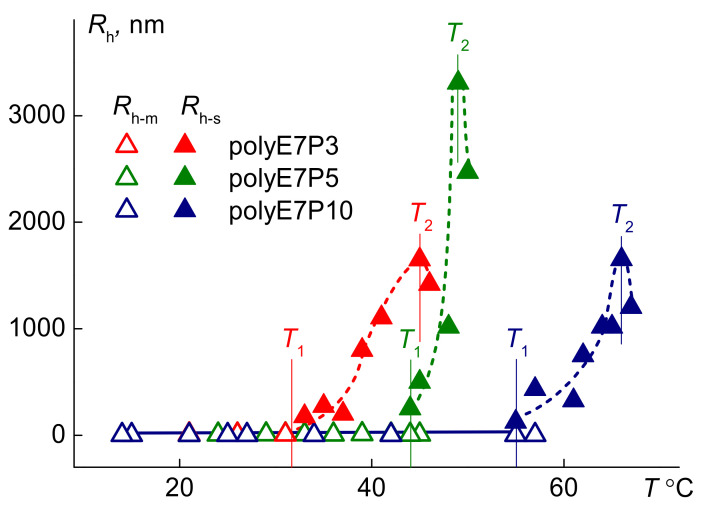
Temperature dependences of the hydrodynamic radii *R*_h-m_ and *R*_h-s_ for the aqueous solutions of polyOEGeOPGpMA at concentration *c* ≈ 0.0050 g·cm^−3^*. I*_21_ is light scattering intensity at 21 °C.

**Table 1 polymers-13-02715-t001:** The molar masses, hydrodynamic characteristics, and CMC of polyOEGeOPGpMAs.

Solvent	*M*_w_, g·mol^−1^	*R*_h-D_, nm	[η], cm^3^·g^−1^	*A*_0_ × 10^10^, erg·K^−1^mol^−1/3^	CMC, wt%
polyE7P3					
water	33,000 ± 3000	3.2 ± 0.3	12 ± 0.6	3.9	3.4∙10^−4^
chloroform	20,000 ± 2000	2.1 ± 0.2	8.0 ± 0.4	4.1	
THF	10,000 ± 1000, 11,000 *	2.1 ± 0.2	6.3 ± 0.3	3.0	
acetonitrile	12,000 ± 1200	2.1 ± 0.2	6.0 ± 0.3	2.9	
polyE7P5					
water	180,000 ± 1800	4.2 ± 0.3	15 ± 0.6	5.2	3.2∙10^−4^
chloroform	50,000 ± 4000	3.9 ± 0.3	9.9 ± 0.5	3.9	
THF	15,000 ± 1500, 14,000 *	2.9 ± 0.3	8.0 ± 0.4	2.7	
acetonitrile	15,000 ± 1500	2.9 ± 0.3	8.0 ± 0.4	2.7	
polyE7P10					
water	77,000 ± 7000	4.3 ± 0.4	19 ± 0.8	4.2	2.7∙10^−4^
chloroform	87,000 ± 8000	4.2 ± 0.4	12 ± 0.6	3.9	
THF	20,000 ± 2000 15,000 *	3.2 ± 0.3	11 ± 0.5	3.0	
acetonitrile	17,000 ± 1700	3.0 ± 0.3	11 ± 0.5	2.8	

* These values of *M*_w_ were obtained by GPC.

**Table 2 polymers-13-02715-t002:** Structural characteristics of polyOEGeOPGpMA.

Sample	*M*_w_, g·mol^−1^	*N* _b_	*L*_b_, nm	*L*_sc_, nm
polyE7P3	~11,000	19	4.8	4.2
polyE7P5	15,000	22	5.5	4.9
polyE7P10	17,000	18	4.6	6.8

## Data Availability

The data presented in this study are available upon request from the corresponding author.

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
