# Peer review of "Conformation, Self-Organization and Thermoresponsibility of Polymethacrylate Molecular Brushes with Oligo(ethylene glycol)-block-oligo(propylene glycol) Side Chains"

_polymers, 2021, doi:10.3390/polym13162715_

Round 1

Reviewer 1 Report

Manuscript Number: polymers-1332167

Title: Conformation, self-organization and thermoresponsibility of polymethacrylate molecular brushes with oligo(ethylene glycol)-block-oligo(propylene glycol) side chains

Article Type: Article

In the manuscript the experimental research concerning thermoresponsibility of polymethacrylate molecular brushes with oligo(ethylene glycol)-block-oligo(propylene glycol) side chains. This is a continuation of previous work of Authors, namely:

Kazantsev, O.A.; Bolshakova, E.A.; Orekhov, D.V.; Simagin, A.S.; Kamorin, D.M.; Sivokhin, A.P. Synthesis and thermoresponsive properties of polymethacrylate molecular brushes with oligo(ethylene glycol)-block-oligo(propylene glycol) side chains. Polymer Bulletin. 2021 (In press)

According to the Web of Science database Authors have vast experience in polymers synthesis and analysis. The manuscript is very interesting. The topic of the research fits perfectly the scope of the Journal. Everything is clearly explained and discussed. In my opinion the biggest advantage of the manuscript is the complete analysis of polymers. On the other hand I have found two disadvantages of the work. Firstly the temperature of the samples was changed in a step-wise manner. It would be very interesting to investigate the impact of heating rate using ramping method on the behavior of samples. However I do understand that the Authors are limited by the capabilities of the apparatus. Secondly, the hysteresis of transmittance could be measured. That would give the full information about the thermoresponibility of samples. Nevertheless I think that the manuscript is interesting and worthy publication.

Below I am presenting my detailed remarks:

  1. The list of symbols together with appropriate SI units used in all equations should be introduced to the manuscript.
  2. Figure 2: Photocor Complex apparatus is able to perform measurements for multiple angles. The angle value may influence the analysis results. Therefore could Authors give information about the angle used in measurements? The measurements were made for one fixed angle or it the graph is an average for multiple angles?
  3. Figure S3 and S4: The presented results concerns extremely sensitive parameters. We are talking about hydrodynamic radius value below 10 nm. Therefore in order to analyze these properties properly the measurements have to be repeated multiple times. Therefore, could Authors be so kind and give information in the manuscript about the number of copolymer samples that were synthesize and the number of independent measurements made for each sample. Moreover the error bars representing the standard deviation of values have to be added to the graphs.
  4. Line 113-116: Please give the temperature at which the measurements were made.
  5. Table 1: As in the case of figures please add the information about the standard deviations of presented values.

I recommend to accept the manuscript after minor revision.

Author Response

The solution temperature T was changed discretely with the step ranging from 1.0 to 5.0 °C. The step was regulating depend on T. Near the phase separation temperatures the step was minimal (1.0 °C) The temperature was regulated with the precision of 0.1 °C. The T values changed in the range from 15 to 79 °C.

The aim of our next work is to study of hysteresis for all samples under different concentration and content of the OPG block in the side chains.

1) Thank you very much for your comments. We corrected manuscript (Page 10).

Photocor Complex apparatus is able to perform measurements for multiple angles (usually in angle range 45-135°). For determination of MM all samples we used Debye plot. The experiments we carried out at angle of scattering θ =90 °. We added this information in text. Page 4

2) We  agree with you. Therefore, in order to analyze value of hydrodynamic radii below 10 nm properly the measurements have to be repeated multiple times. For all samples (copolymer samples were 3) in all solvents (4 solvents) were carried out several experiments (independent measurements made for each sample about 10 times). Moreover, we gave distribution of radii (several times) until the value of Rh was constant. Note, that the studied samples had low degree of dispersity.

3) Standard deviation of values hydrodynamic radii have added in figures and text (table 1) of paper.

4) Measurements were made at 25°C. We added this information in paper, page 3.

5) We added this information in figures, table, and manuscript. Figures S3 and S4 (Supplementary materials) 

List of symboles and figures in added files (word)

Reviewer 2 Report

The paper of Simonova et al. reports a study of the effect of oligo (ethylene glycol) and oligo (propylene glycol) block lengths in polymethacrylic molecular brushes leading oligo(ethylene glycol)-block-oligo(propylene glycol) side chains. Polymers were investigated by static and dynamic light scattering and viscometry in different solvents. The article is well written and The authors have done a good job of characterization of co-polymers and results are clearly presented. Some English and typing errors are present in the manuscript. They should be corrected. Quality of Figure 1 should be improved. Therefore, I can recommend the paper for publication on Polymers after minor revision.

Author Response

Thank you very much for your comments. We corrected manuscript and improved of quality of Figure 1. Page 2.
